# Bio-Based Polyurethane Composites and Hybrid Composites Containing a New Type of Bio-Polyol and Addition of Natural and Synthetic Fibers

**DOI:** 10.3390/ma13092028

**Published:** 2020-04-26

**Authors:** Adam Olszewski, Paulina Kosmela, Aleksandra Mielewczyk-Gryń, Łukasz Piszczyk

**Affiliations:** 1Department of Polymer Technology, Chemical Faculty, G. Narutowicza St. 11/12, Gdansk University of Technology, 80-233 Gdansk, Poland; Adam.olszewski@pg.edu.pl (A.O.); paulina.kosmela@pg.edu.pl (P.K.); 2Faculty of Applied Physics and Mathematics, Gdansk University of Technology, Narutowicza St. 11/12, 80-233 Gdansk, Poland; alegryn@pg.edu.pl

**Keywords:** polymer matrix composites (PMCs), hybrid composites, glass fiber, sisal fiber, biomass liquefaction, scanning electron microscopy (SEM)

## Abstract

This article describes how new bio-based polyol during the liquefaction process can be obtained. Selected polyol was tested in the production of polyurethane resins. Moreover, this research describes the process of manufacturing polyurethane materials and the impact of two different types of fibers—synthetic and natural (glass and sisal fibers)—on the properties of composites. The best properties were achieved at a reaction temperature of 150 °C and a time of 6 h. The hydroxyl number of bio-based polyol was 475 mg KOH/g. Composites were obtained by hot pressing for 15 min at 100 °C and under a pressure of 10 MPa. Conducted researches show the improvement of flexural strength, impact strength, hardness, an increase of storage modulus of obtained materials, and an increase of glass transition temperature of hard segments with an increasing amount of fibers. SEM analysis determined better adhesion of sisal fiber to the matrix and presence of cracks, holes, and voids inside the structure of composites.

## 1. Introduction

Studies on polymer composites and the application of various renewable resources and fillers in production of polymer matrix composites have been one of the most popular areas of active research for the past several years [1,2,3,4,5]. The most commonly used fillers are glass fiber, carbon fiber, carbon nanotubes (CNTs), and finally, natural fibers, such as kenaf, sisal, bamboo fiber, and others [6,7,8,9,10]. Glass fibers are used often in the preparation of polymer composites. It has been reported that mechanical properties such as tensile strength, Young modulus, and compression modulus increase at the right content. In the abovementioned studies, these properties were examined in a variety of glass fiber polymer composites, where polymer matrix was containing polyester resin, epoxy resin [10], polyurethane [11], or polyolefin [12].

The growth of the environmental awareness, global warming, and preserving nature has attracted attention to an examination of natural fiber properties. This process resulted in the expansion of the usage of natural fibers and replacing the synthetic ones in various applications mainly in construction, as well as in the automotive and other industries [13]. Many studies confirmed that the fabrication of biodegradable composites by adding natural lignocellulose fibers into synthetic polymer matrices may cause a less harmful effect on the environment than petroleum-based materials. Furthermore, non-degradable synthetic plastics increase the dependence on petroleum-based products. Unfortunately, processing thermal and mechanical properties of these reinforcements can cause a lot of problems making them undesirable for certain applications [6,14]. Moreover, natural fiber-reinforced composites have few disadvantages, such as poor compatibility with some matrices, poor wettability, and high moisture absorption of fibers.

The most popular hybrid composites are glass fibers with kenaf, sisal, banana, jute, date palm wood, basalt flax, coir, and oil palm fiber empty fruit bunch [5,15]. One of the most promising natural plants is sisal due to a short plantation time and easy cultivation. About 4.5 million tons of sisal fiber is obtained all over the world each year. The sisal fiber is extracted from sisal plant leaves [15]. This plant is harvested in a few countries in Africa, the Far East, and in West India [16]. The chemical composition of this fiber can be different depending on the location and time of harvesting. This material consists of lignin, cellulose, hemicellulose, and moisture. Comparing the mechanical properties of sisal fiber to the glass fiber it can be noticed that these two fibers have an almost similar mechanical specification [15,17,18]. Sisal fiber is one of the most popular natural fibers used in the industry due to its low price, high availability, good tensile strength, high toughness, good thermal, and acoustic insulation properties. Furthermore, sisal fibers have been used as a reinforcement element in composites in the past few years [19].

These fibers can be used as reinforcement in polyurethane resins (PU) which are synthetic polymers that have the urethane groups in their structure (–NHCOO–). The PU resins can be prepared by a reaction of polyols, water, and amines containing an active hydrogen atom with the NCO group contenting isocyanate. This material contains two types of segments in their molecular chains—hard segments and soft segments [6,13]. The first of them is mainly composed of the isocyanate structure and part of the chain extender which can aggregate between the molecular chains due to the hydrogen bonds, which can provide the rigidity and high mechanical properties of PU resins. The soft segments are composed of a long chain structure of the polyol, which can provide the PU resins elasticity. The biggest advantage of polyurethanes is the possibility of the design properties of a material to satisfy different requirements. Different types and proportions of polyols, isocyanates, and additives (flame retardants, dyes, catalyst, and other) can affect the properties of the final product. Moreover, the processing conditions, such as temperature, method of processing, and time, have a huge impact on the structure of the PU resins. Due to their properties and outstanding molecular design, these materials have a huge variety of practical applications, such as adhesives, foams, plastics, and rubber. The molecular structures can have many forms from linear, highly extensible elastomers to rigid crosslinked polymers. The curing of PUs involves the formation of a three-dimensional network through reactions of polyfunctional molecules [5,20]. This process starts from the growth of the linear polymer chain. Next, the chain starts to branch and after that crosslinking begins. During curing the molecular weight increases sharply and the molecular size increases. After this process, chains are linked together into a network. The most commonly used isocyanates are methylene diphenyl diisocyanate (MDI), toluene diisocyanate (TDI), and hexamethylene diisocyanate (HDI). Polyurethanes based on MDI and TDI are widely used because of their excellent mechanical properties [5].

The second most important raw material in the production of polyurethanes are polyols, which are polyhydric compounds divided into polyester and polyether polyols [10,21,22,23,24]. In the literature, there is more and more information about polyols derived from vegetable oils. These raw materials in terms of chemical structure are esters of glycerol and unsaturated fatty acids [22,23]. The general interest in obtaining polyols from renewable raw materials is caused by shrinking oil resources and the constant increase in gas and oil prices. One of the methods of obtaining bio-based polyols is liquefaction. This method relies on turning whole biomass into liquids by a reaction of biomass with a suitable solvent and catalyst. Different lignocellulosic materials can be used in the liquefaction process. The most popular raw biomass materials are agricultural crops, energy crops, aquatic plants, bamboo, and woody plants. Moreover, liquefaction is a process which allows the transformation of different wastes (animal, agricultural, forestry, industrial, food) into useful products [25,26,27]. Lignocellulosic biomass is composed of cellulose, hemicelluloses, lignin, and a small number of other substances. It has multiple methods of conversion, various end-products, and wide application areas [25]. An application in the industry of sustainable and renewable biomass can solve the problem of lack of materials and reduce the consumption of non-renewable fossil fuels. The process of the liquefaction can be influenced by changing chemical compounds or physical factors. The most important chemical factors are the type and size of biomass, type of used solvent, and catalysts. Physical factors include temperature, pressure, component mass ratio, catalyst concentration, and heating rate [25,26,28]. 

Types of catalyst and solvent take essential roles in the liquefaction process. Homogeneous catalysts (organic and inorganic alkalis, acids, and salts) are more active and popular than heterogeneous catalysts. Studied catalysts including acidic catalysts (hydrochloric acid, sulfuric acid, p-toluenesulfonic acid, oxalic acid), metallic salts, alkalis (sodium hydroxide), and acidic ionic confirmed that using strong acids as catalyst in the liquefaction process can cause corrosion of metallic equipment and environmental pollution compared with an alkali catalyst, but promotes higher efficiency of liquefaction [21,25,29].

In the present research, cellulose biomass liquefaction with the use of crude glycerol and a polyethylene glycol mixture as a solvent in addition to sulphuric acid as a catalyst was carried out. The obtained bio-based polyols were characterized. Moreover, eight different composites and hybrid composites were prepared. Properties of fibers, polyols, and obtained materials were characterized using a variety of research methods, such as hydroxyl number, biomass conversion, flexural and impact strength, Fourier-transform infrared spectroscopy, dynamic mechanical analysis, and scanning electron microscopy.

## 2. Materials and Methods

### 2.1. Materials

Substrates to obtain bio-based polyol PEG 400 (polyethylene glycol) were supplied by POCH S.A (Gliwice, Poland) and refined crude glycerol was acquired from Bio-Chem sp. z o.o. (Olszanka, Poland). As a catalyst, 95% aqueous solution of sulfuric (VI) acid was purchased from Avantor Performance Materials Poland S.A (Gliwice, Poland). Shredded cellulose was obtained from POCH S.A (Gliwice, Poland). Polyether polyol Rokopol^®^ M6000 (L_OH_ = 28 mg KOH/g) was supplied by PCC Rokita S.A (Brzeg Dolny, Poland). The polymeric methylene diphenyl diisocyanate (pMDI) was supplied by Borsodchem (Kazincbarcika, Hungary). The content of free isocyanate groups in pMDI was 31.5%. Dibutyltin dilaurate 95% was obtained from Sigma-Aldrich (Saint Louis, MO, USA). Glass fiber of 10 mm was purchased from KROSGLASS S.A (Krosno, Poland), and sisal fiber was obtained from Canella S.A (New York, NY, USA).

### 2.2. Preparation of Bio-Based Polyols

Three bio-based polyols were synthesized by the liquefaction process of shredded cellulose. PEG 400 and refined crude glycerol were used as a solvent in this reaction. As a catalyst, 95% water solution of sulfuric acid was used. The mass ratio of cellulose to the solvent was 1:10. The catalyst (3 wt.%) was added to the reactor to accelerate the process. The reaction was run for 6 h at 120 °C, 150 °C, and 180 °C to determine the impact of temperature on the reaction. Samples were collected every hour. After this process, the obtained polyol was neutralized with sodium hydroxide pellets and dried under vacuum for 2 h. Figure 1 shows the main cellulose liquefaction reaction. Polyol with the best properties (appropriate hydroxyl number, biomass conversion, viscosity, and water content) was selected for further research.

### 2.3. Preparation of Polyurethane Composites and Hybrids

Glass/sisal fibers were mixed with polyols (90% of obtained bio-polyol and 10% of M6000), pMDI, and dibutyltin dilaurate in a beaker and stirred until a homogeneous mixture was obtained. Next, the mixture was poured into the mold and hot pressed for 15 min at 100 °C under a pressure of 10 MPa. Eight sets of composites of different ratios of sisal and glass fiber were fabricated as shown in Table 1.

### 2.4. Characterization

The hydroxyl number of bio-based polyol was determined according to PN-EN 1240:2011. Samples of 0.5 g were placed in a 250 cm^3^ Erlenmeyer flasks with the acetylating mixture. Samples were heated for 30 min, then 1 mL of pyridine and 50 mL of distilled water were added. Finally, the obtained mixtures were titrated using 0.5 M KOH solution in the presence of phenolphthalein until the color of the mixture changed to pink.

The water content of the produced bio-based polyols was assessed by using Karl Fischer titration. The samples were diluted with methanol and titrated with Fischer reagent. Biomass conversion was evaluated on the basis of the percentage of residual biomass. The liquefaction product was diluted with methanol, stirred with a magnetic stirrer for 1 h, and then filtered through the filter paper under vacuum. The solid residue was washed with methanol and then dried in the oven to a constant weight.

FT-IR (Fourier-transform infrared spectroscopy) spectrophotometric analysis was performed in order to determine the structure of the cellulose, the bio-based polyol, fillers, and obtained polyurethanes. The analysis was performed at a resolution of 4 cm^−1^ using a Nicolet 8700 apparatus (Thermo Electron Corporation, Waltham, MA, USA) equipped with a Snap-Gold State II which allows measurements in the reflection configuration mode.

The rheological measurements allowed the characterization of rheological behavior and mathematical model of obtained polyol. Rheological measurements with the controlled shear rate were performed by using a rotary R/S Portable rheometer. Tests were performed at 30 and 50 °C. The shear rate was increased from 1 to 100 s^−1^ for 120 s and then decreased to a minimum. Based on the measurement, the viscosity and flow curves of obtained polyol were plotted. The rheological parameters were calculated and the rheological models were defined by using the Rheo3000 program.

The flexural tests were performed with rectangular specimens in accordance with PN-EN ISO 14125. The samples with normed dimensions were measured with a slide caliper with an accuracy of 0.1 mm. The tests were performed on a Zwick/Roell 1000 N testing machine at a constant speed of 100 mm/min until fracture. The maximum load was applied in the middle of the specimen. The specimen was freely supported. 

The impact strength was performed using Izod type hammer. The energy of the hammer was 5.5 J. The test was executed according to PN-EN ISO 180:2004 and samples were beam-shaped. The hardness of obtained composites was examined using the Shore method. The test was executed according to PN-EN ISO 868:2005.

The dynamic mechanical analysis was performed using the DMA Q800 TA Instruments apparatus. Samples were analyzed in strain mode with a frequency of 1 Hz. Measurements were performed at a temperature range from −120 to 140 °C with a heating rate of 4 °C/min. Samples were beam-shaped and had dimensions of 10 mm × 40 mm.

Morphological investigations were performed on the prepared hybrid composites with SEM Quanta FEG 250. Before the analysis, samples were coated with a 10 nm gold layer. The observations were carried out on the samples subjected to the brittle cracking in the liquid nitrogen.

## 3. Results and Discussion

### 3.1. Properties of Fibers, Cellulose, and Obtained Polyols

The variation of the hydroxyl number over time is presented in Figure 2. The reaction at 170 °C was carried out only for 4 hours because of the high viscosity of the system. For the temperature of 120 °C and reaction time 6 h, the hydroxyl number of bio-based polyol was about 600 mg KOH/g. For polyol synthesized at 150 °C, the hydroxyl number of was 475 mg KOH/g, which is comparable to the commercially available polyols.

From a technological point of view, the hydroxyl number is a very important parameter for the production of polyurethane materials. The hydroxyl value of obtained polyol depends on reaction time and temperature. During synthesis of polyol, reactions are occurring between hydroxyl groups of glycerol particles, hydroxyl groups of condensed glycerol or PEG particles, and groups present in cellulose [22,30]. The total amount of hydroxyl groups is decreasing during the synthesis and hence a drop of the hydroxyl number can be observed. This has also been reported by others [1,30]. After two hours at 150 °C, the total amount of hydroxyl groups increased. This can be associated with the generation of side products during the reaction and decomposition of the long chains of biomass [22]. During the reaction, there is a possibility that excess solvent may condensate to oligomers, polymers, and generation of water particles. These water particles can decompose the structure of biomass and lead to the generation of hydroxyl groups. The presence of this group may increase the hydroxyl number of polyols [31]. After the neutralization process, the hydroxyl number has increased slightly because of the generation of water during the reaction between KOH and H_2_SO_4_. Polyol synthesized at 150 °C had the lowest water content (0.27%), which shows the high efficiency of the drying process.

In Figure 3, the changes in biomass conversion during the reaction are shown. During liquefaction, biomass undergoes degradation via decomposition of long biomass chains to low molecular weight compounds, which can react with the solvent. This process results in higher biomass conversion. Degradation using sulphuric acid determines the first stage of liquefaction, leading to high process efficiency. During synthesis, the degradation products can undergo repolymerization, which causes the presence of insoluble material composed of glycol or glycerol glucosides and xylosides. The presence of these compounds decreases the rate of biomass conversion [1,32]. The decrease of the hydroxyl number and the increase of biomass conversion at the same time may indicate a higher extent of conversion, which is described in the literature [31,33].

The fastest biomass conversion was observed for the temperature of 170 °C; however, the too high viscosity of the system does not allow its use in further researches. At 120 °C, a satisfactory level of conversion was not achieved, which resulted in the products obtained having inferior properties. The best properties (low water content, suitable viscosity, and high biomass conversion) were obtained for a reaction temperature of 150 °C and a reaction time of 6 h.

The influence of temperature and shear rate on viscosity plays an important role from a technological point of view. The determination of the character of liquids allows for a correct adjustment of processing conditions and apparatus such as mixers during further industrial production. According to the obtained results, the dynamic viscosity and flow curves are presented in Figure 4 and Figure 5.

The results show that obtained bio-based polyol shows the linear flow and can be described as a Newtonian fluid. According to obtain the curves (Figure 4.), it was noticed that shear stress decreases with increasing temperature. This effect is caused by the higher mobility of macromolecules and free volume between them. The maximum value of shear stress was 198.6 Pa and it was observed at 30 °C. This parameter decreased to 52.2 Pa when the temperature was raised to 50 °C. The bio-polyol obtained in the liquefaction process is a rheologically stable fluid because the deviation of the flow curve hysteresis is small.

Figure 6 shows the FTIR spectra of glass and sisal fibers. For the sisal fiber, the peak near 3300 cm^−1^ confirms that hydroxyl groups (OH) are present in the sisal fiber. Due to the presence of these groups in the structure of sisal fiber, it can be considered as a reactive filler. The OH groups in the fibers can absorb moisture and form a weak interface when incorporated into a polymer matrix. The vibrations near 1730 cm^−1^ and 1250 cm^−1^ can be associated with C=O and C–O bonds in cellulose, lignin, and hemicelluloses, which are present in the sisal fiber structure. In addition, a C–H vibration peak at 2910 cm^−1^ can be observed. The signal near 1410 cm^−1^ can be related to the stretching of the CH_2_ groups [27,34,35,36,37] For the glass fiber, the most important peak occurs about 900 cm^−1^ and can be related to the Si–O–Si bonding.

Figure 7 shows FT-IR spectra of cellulose (biomass), polyol, and matrix. For the polyol the characteristic OH-stretching vibrations band is visible at 3400–3300 cm^−1^. The band visible at 2950–2860 cm^−1^ can be assigned to symmetrical and asymmetrical CH-stretching vibrations originating from the CH_2_ groups present in aliphatic chains, or the CH_3_ groups. The band at 1720 cm^−1^ can be related to CO-stretching vibrations in cellulose or to the vibrations caused by carbonyls generated during cellulose degradation at high temperatures. The absorption bands at 1440–1340 cm^−1^ have been assigned to CH_2_ and HOC in-plane bending vibrations in cellulose resulting from aromatic skeletal vibrations combined with a symmetrical mode of bending hydroxymethyl. The signal at 1210 cm^−1^ can be attributed to C–O and C=O-stretching vibrations. The characteristic for the ether group vibrations can be spotted in the range of 1170–980 cm^−1^. C–H bonds that are present in the cellulose structure and C=C bond can be related to the signals at 920–850 cm^−1^ [1,33,34,38,39].

### 3.2. Properties of Obtained Matrix, Composites, and Hybrids

The Fourier-transform infrared spectroscopy of the sample without the addition of filler is presented in Figure 7. A signal characteristic for stretching vibrations of the N–H groups (in urethane linkages) was observed in the range 3280–3300 cm^−1^. Bands attributed to the bending vibrations of these groups were observed at 1510–1520 cm^−1^. The absorption bands at 1700–1720 cm^−1^ can be assigned to CO-stretching vibrations in cellulose or to the vibrations caused by carbonyls generated during cellulose degradation at high temperatures. Signals at 1530 cm^−1^ can be associated with stretching vibrations of C–N bonds in urethane linkages, and the bending vibrations of C–N were registered at 1599 cm^−1^ [32,33]. Signals at 2260–2280 cm^−1^ can be assigned to the presence of unreacted N=C=O groups. Absorption bands mentioned above confirm the presence of urethane bonds in the investigated material. The signals at 2950–2860 cm^−1^ were attributable to the symmetric and asymmetric stretching vibrations of CH bonds in CH_2_ groups present in aliphatic chains and CH_3_ end groups, while the deformation vibrations were observed at about 1420 cm^−1^. Bands in the range of 1170–850 cm^−1^ are related to the structure of the obtained polyols [1,34,38,39].

Table 2 shows flexural strength, maximal deformation, impact strength, and hardness of obtained composites. Glass and sisal fibers as a reinforcing agent were investigated. It is shown that both of the fibers reduce the deformation at fracture of prepared samples. The earlier cracking of the samples can be caused by increased stiffness of composites, stress concentration, and possible perpendicular position of fibers to the bending axis. The deformation of composites is reduced from 12.62% to 8.9% and to 5.27% for 15%GF and 15%SISAL, respectively. In comparison to glass fiber, the content of the sisal fiber has a higher impact on the elasticity of obtained composites and faster reduces the maximal deformation of samples. This effect can be caused by the weak compatibility between matrix and glass fiber (GF), as well as a higher crosslinking density of sisal fiber composites, which is caused by the reaction of OH groups present in the sisal fiber with isocyanate. Moreover, it may be due to the crack propagation across two phase boundaries (fiber/matrix) in the composites [21]. Moreover, the addition of sisal fiber affects the isocyanate index, which change the properties of the resulting composites. On the other hand, the flexural strength of the composites filled by 15% of sisal and glass fiber has been greatly improved. The flexural strength of composites filled with GF and sisal has improved to respectively 24.43 MPa and 25.72 MPa. The biggest impact on the properties can be noticed in the hybrid composites with both fillers. For sample 10%GF/5%SISAL the flexural strength has been improved from 16.41 MPa to 43.04 MPa [40,41].

The impact strength of the obtained composites is higher than the matrix. This confirms the strengthening effect of both fillers. The impact strength of composites is higher for samples with a higher content of fillers, which increases for 15%GF and 15%SISAL from 0.135 to, respectively, 1.452 and 0.914 kJ/m^2^. Again, the biggest impact on the properties can be noticed in the hybrid composites with both fillers. For sample 10%GF5%SISAL the flexural strength has been improved to 1.699kJ/m^2^. This effect can be caused by the transmission of stress by fibers, fracture mechanism modification, and limiting the motion of matrix polymer chain [40,41].

As shown in the table, the increase of the hardness can be observed with the presence of the fibers. Sisal fiber has a stronger impact on the hardness of composites and this parameter has been increased by 24%. For the combination of both fibers, the hardness has been increased by 33%. It was observed that the increase of hardness and stiffness is associated with a decrease of deformability. This test confirms the strengthening effect of this combination [42,43].

The variation of the storage modulus (E’) is shown in Figure 8 and Figure 9. DMA was performed to examine how the exposition of the hybrid composites to elevated temperatures would affect the properties of the composites. The change of the storage modulus occurs with the type and amount of fillers. It was found that the addition of both fillers increased the modulus in a wide range of temperatures. Values of the storage modulus have increased from 584 MPa for matrix to, respectively, 2369 MPa and 2222 MPa for 15%GF and 10%GF5%SISAL. In contrast to sisal, it was observed that samples with a higher content of glass fiber have a higher storage modulus. Samples of hybrid composites have properties between 15%GF and 15%SISAL and the storage modulus decreases with a lower amount of glass fiber. Furthermore, in higher temperatures (T > 60 °C), composites with sisal fiber have higher value of E’. It can be due to bonding and interactions between sisal fiber and polymer matrices. If the bonding between polymer matrix and fiber is stronger and the fiber modulus is higher, the increase of E’ can be observed. In the literature it is described that the modulus in a glassy state results from the intermolecular forces between atoms and the internal structure of polymer. That explains the reinforcing effect of using fibers, which allows the transfer of stress to the fiber [6,44].

The variation of the loss modulus is shown in Figure 10 and Figure 11. It was noticed that the increase of storage modulus is associated with the decrease of loss modulus. During the analysis of the curves, the presence of two phases has been observed. The first phase is made of soft segments of bio-polyol and the second one is made of hard segments of isocyanate. It was found that the addition of both fillers also increases the loss modulus of composites. The highest values of the E’, at 293,7 MPa and 164,8 MPa, revealed samples 15%GF and 10%GF5%SISAL, respectively. Again, the samples of hybrid composites have properties between 15%GF and 15%SISAL, and the loss modulus decreases with a lower amount of glass fiber. Additionally, using values of the loss modulus, the temperatures of glass transition (T_g_) of both segments have been summarized in Table 3. It was observed that the addition of both fibers does not affect significantly the glass transition temperature of soft segments. On the other hand, T_g_ of hard segments increases with the addition of fiber, especially for the hybrid composites. This effect can be associated to the decrease of the polymer chain mobility in the matrix. An increase of T_g_ can be caused by the interactions between the polymer and the sisal fibers due to the existence of OH groups on the PU surface and on the surface of the sisal fibers, as was proved by the FTIR analysis.

Figure 12 and Figure 13 show a variation of the loss factor (Tanδ), which is the ratio between storage and loss modulus. Tanδ gives information about interactions between fiber and polymer matrix [45]. It was noticed that damping factors of samples filled with glass fiber and 5%SISAL are higher than the matrix. The value of this parameter for 15%GF, 15%SISAL, and matrix is respectively: 0.35, 0.15, and 0.23. The increase of this parameter can be associated with the restricted mobility of molecular chains of the system and thus less energy must be consumed to overcome the friction between molecular chains. Due to good interactions between fiber and matrix reduction of the damping factor for sample 15%SISAL can be observed [6,21].

Sisal and glass fiber micrographs are shown in Figure 14 and Figure 15. In Figure 14 we can observe that short cells are joined together along the sisal fiber. Moreover, the surface roughness of the sisal fiber can be noticed. The presence of these cells and high roughness increases the fiber surface, which can increase the interaction surface and the fiber’s adhesion to the matrix. Holes inside the sisal fiber can be noticed. Figure 15 shows compact glass fiber packs. Due to absence of characteristic functional groups and plain surface, which limits the interaction surface, adhesion of glass fiber to the matrix can be reduced.

Figure 16, Figure 17 and Figure 18 show micrographs of fractured samples. After the break, the fiber surface is very rough with no regular shape. In all figures, it can be seen that only fracture of sisal fiber and the matrix occurred. The presence of matrix adhered to the fiber structure and material of matrix in sisal fiber holes can be observed. This effect indicates a strong adhesion between the sisal and the matrix. This is also confirmed by transverse and longitudinal cracks in the fiber shown in Figure 16. No evidence of a sisal fiber pull-out can be noticed. Unlike the sisal fiber, the glass fiber did not rupture during breaking of the sample. It shows that the crack propagation in this type of composite could occur on the phase border (fiber/matrix). This can be due to high tensile strength and the plain structure of the glass fiber, which caused insufficient interactions (adhesion) between matrix and glass fiber. Poor contact and bonding between this fiber and matrix resulted in poor stress transfer between phases. For this reason, fiber pull-off occurred and holes can be observed. Moreover, weak interactions between the polymer matrix and the fiber causes discontinuity in the composite structure, as a result of which microcracks are formed which are responsible for further propagation of the fiber’s cracks.

In Figure 16 and Figure 17, voids, holes, and bubbles can be observed. The bubbles are formed mainly due to introducing air into the material during mixing, presence of water in the polyol (generation of carbon dioxide during the reaction between water and isocyanate), and hydrophilic character of the natural fiber. The presence of agglomerates of fibers, voids, and bubbles could affect tensile properties of obtained materials. In Figure 18, it can be noticed that the matrix material adheres better to the surface of the sisal fiber than to the glass fiber. It can be caused by differences of strength of interactions between matrix and both fibers.

## 4. Conclusions

In this research, three different bio-based polyols were synthesized from the biomass—shredded cellulose. Selected bio-based polyol has been used in the preparation of materials. Materials were prepared by hot pressing at a temperature of 100 °C. It has been confirmed that glass and sisal fibers increase the mechanical properties of materials. This research showed a possibility for the future use of bio-based polyols in the manufacturing of functional materials. Obtained polyol has a hydroxyl value of 475 mg KOH/g, which is comparable to the commercially available polyols and can be used as a substitution of petrochemical polyol. Additionally, a high degree of biomass conversion was obtained. Rheological studies have shown that the obtained polyol has the characteristics of Newtonian fluid and shear stress decreases with increasing temperature. The results confirm the future use of the obtained polyol and materials in the industry. FTIR was performed to characterize the fibers, obtained polyols, and materials. It allowed to determine the presence of characteristic functional groups. It has been shown that sisal fiber has an OH group in its structure and can therefore be considered as a reactive filler and probably can guarantee greater interaction between matrix and material. The flexural strength, impact strength, and hardness of materials have been improved by the addition of glass and sisal fibers. These tests proved a strengthening effect of used fibers in the composites. DMA results showed that the storage modulus of composites and hybrid composites increased. The addition of both fibers did not affect the glass temperature of soft segments but increased the temperature of the glass transition of hard segments. T_g_ of hybrid composites has been increased more than T_g_ of normal composites. The change of Tanδ value for samples can be associated with the mobility limitation of molecular chains of the system and decrease of energy consumption to overcome the friction between molecular chains. The reduction of the damping factor for selected samples shows the existence of interactions between fiber and matrix. SEM analysis has determined that the sisal fiber has better adhesion to polymer matrix. This may be due to the presence of hydroxyl groups in cellulose/lignin, which is contained in sisal. The presence of broken sisal fibers, voids, and fiber breaks was noted.

## Figures and Tables

**Figure 1 materials-13-02028-f001:**
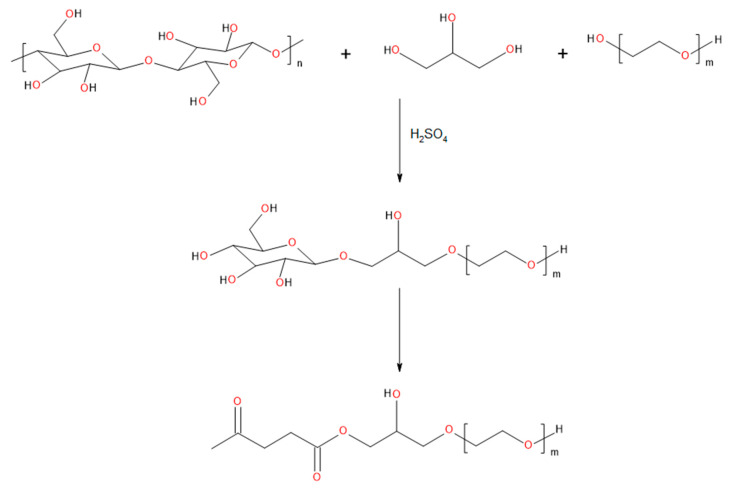
Proposed course of the main reaction occurring as a result of liquefaction of cellulose biomass.

**Figure 2 materials-13-02028-f002:**
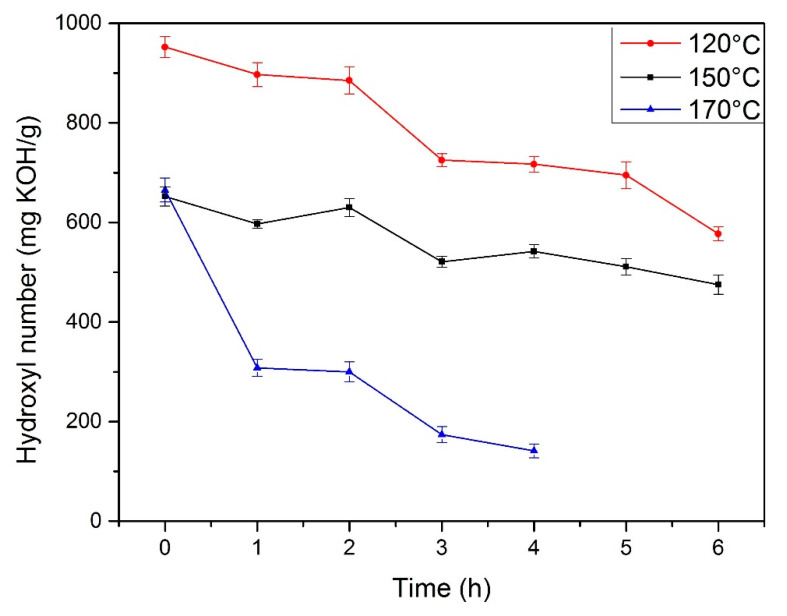
Change of hydroxyl number of obtained polyols.

**Figure 3 materials-13-02028-f003:**
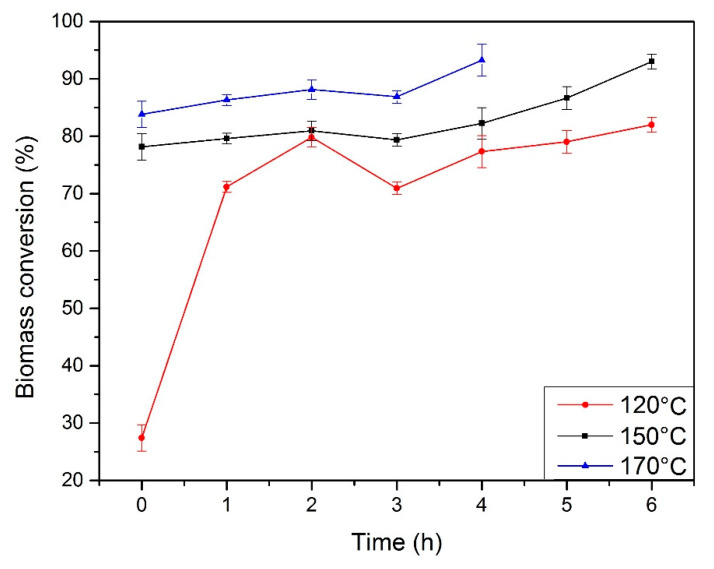
Change of biomass conversion of polyols.

**Figure 4 materials-13-02028-f004:**
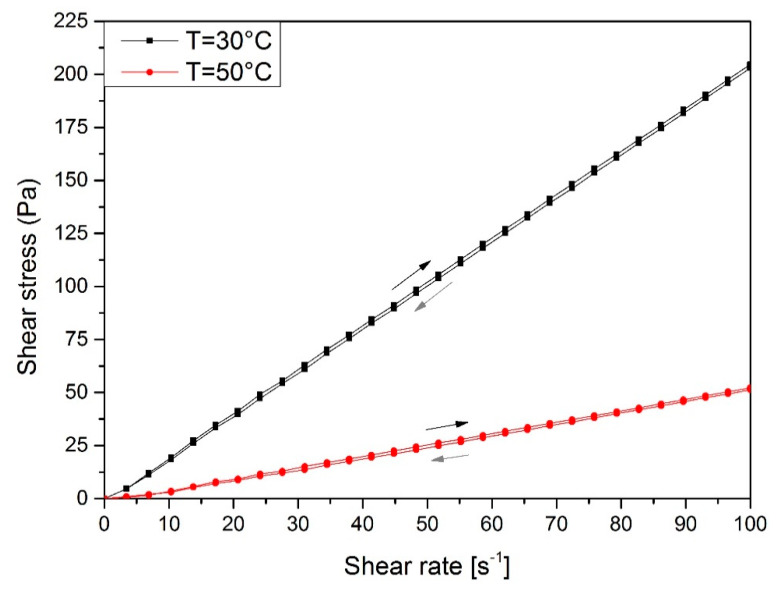
The flow curves of obtained polyol at different temperatures.

**Figure 5 materials-13-02028-f005:**
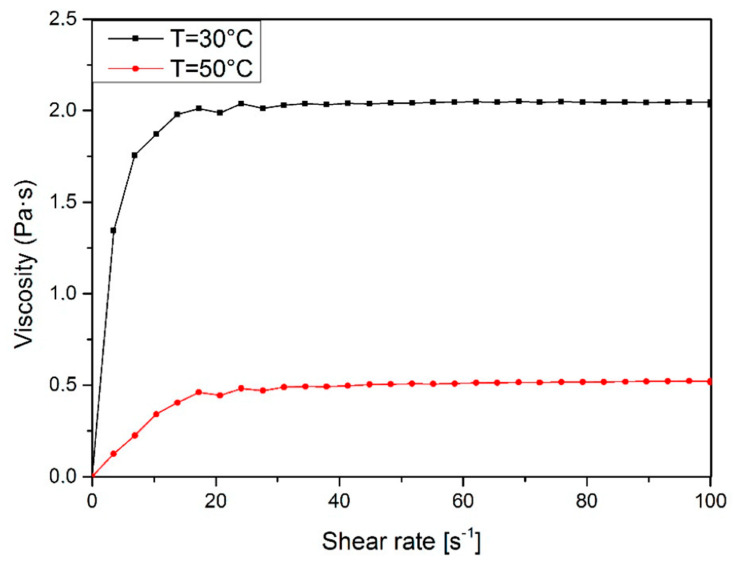
The viscosity curves of obtained polyol at different temperatures.

**Figure 6 materials-13-02028-f006:**
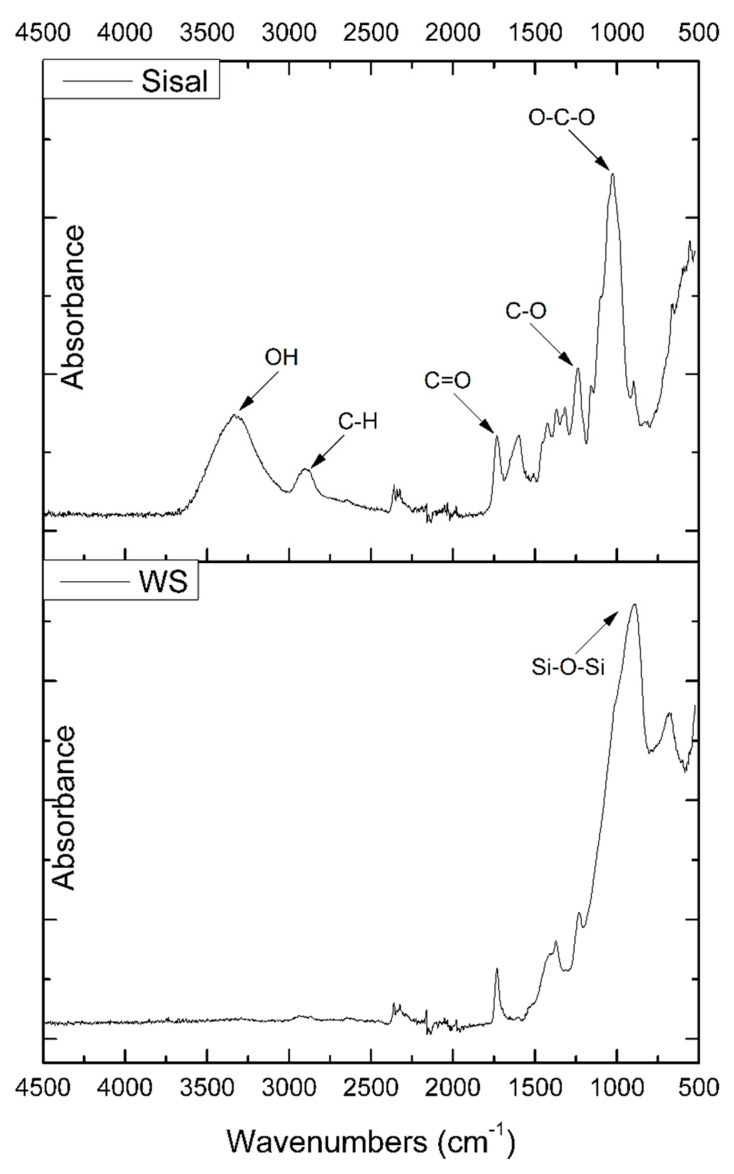
FTIR spectra of sisal and glass fiber (GF).

**Figure 7 materials-13-02028-f007:**
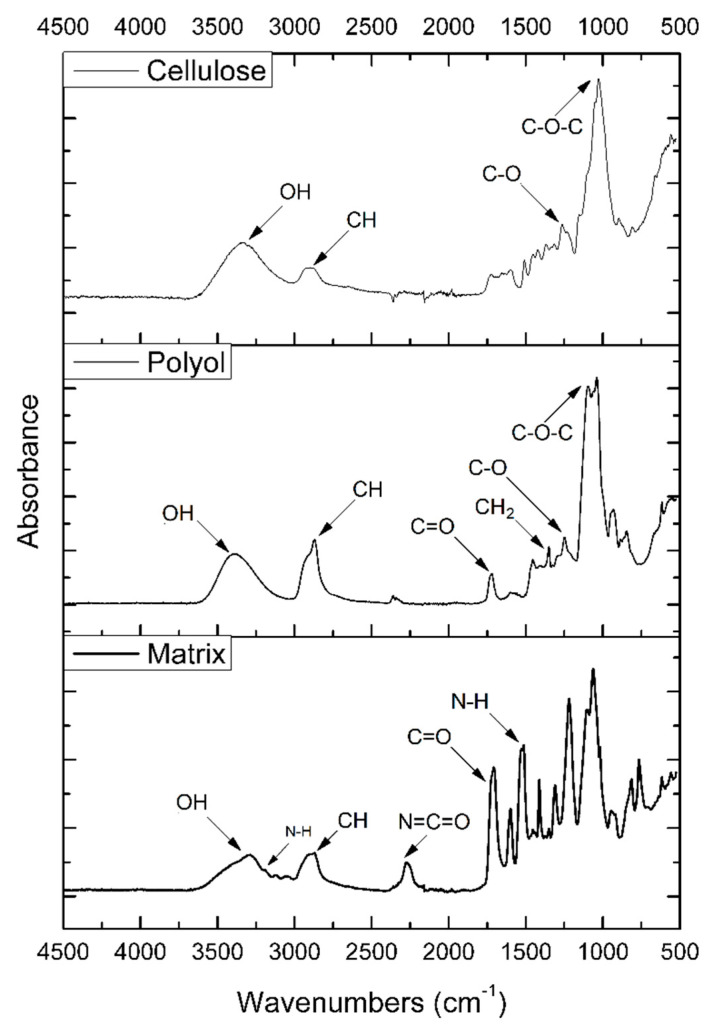
FT-IR spectra of biomass (cellulose), polyol and matrix.

**Figure 8 materials-13-02028-f008:**
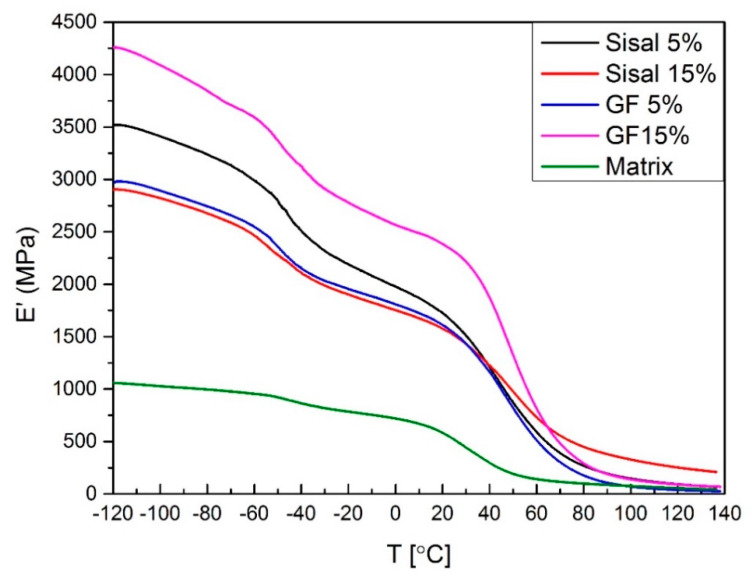
Storage modulus values of sisal and GF composites.

**Figure 9 materials-13-02028-f009:**
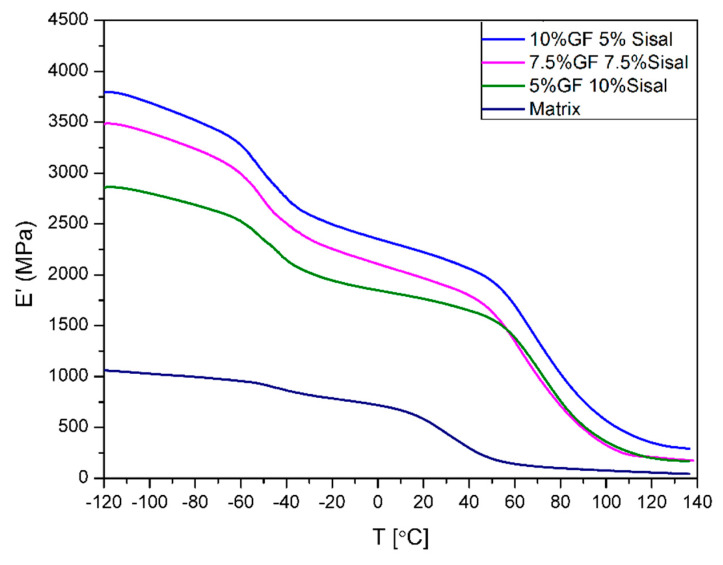
Storage modulus values of obtained hybrids.

**Figure 10 materials-13-02028-f010:**
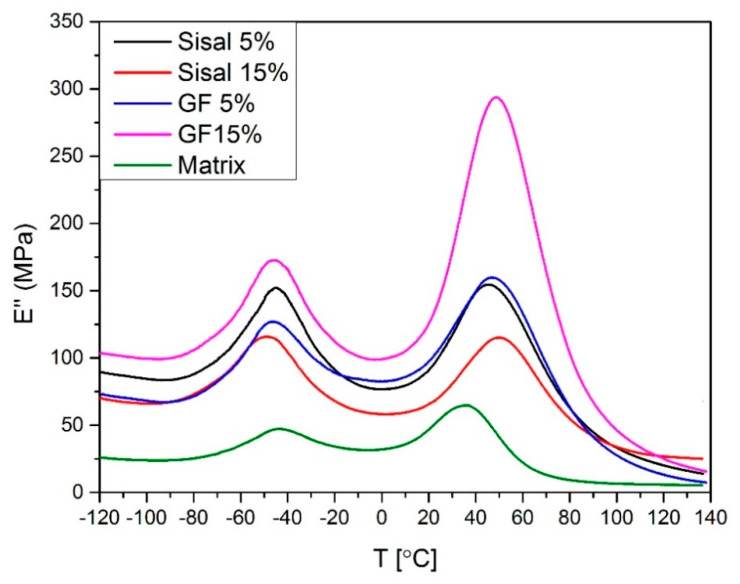
Loss modulus values of obtained hybrids.

**Figure 11 materials-13-02028-f011:**
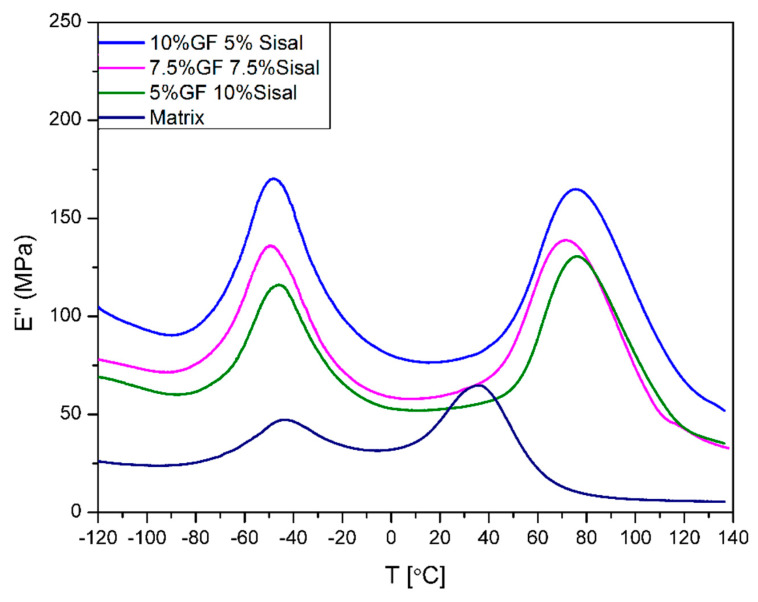
Loss modulus values of obtained hybrids.

**Figure 12 materials-13-02028-f012:**
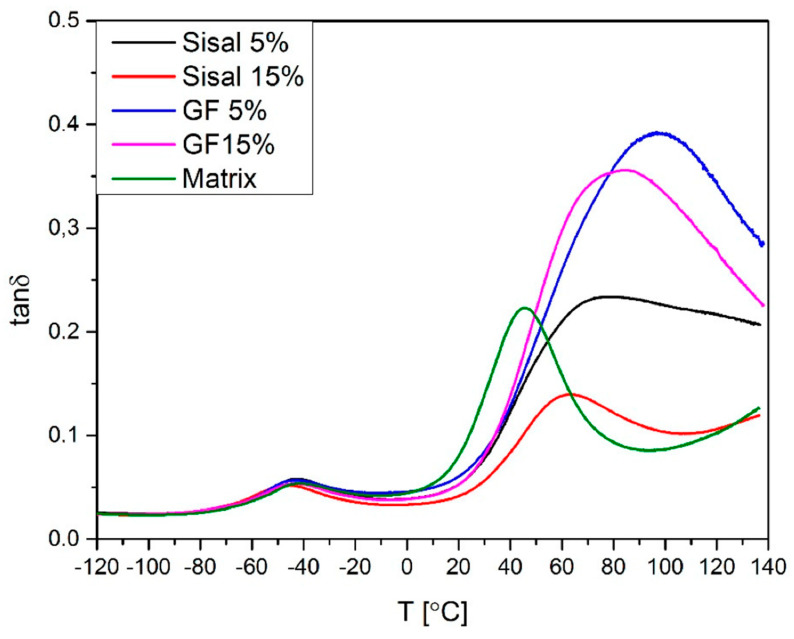
Loss factor (Tanδ) values of obtained composites.

**Figure 13 materials-13-02028-f013:**
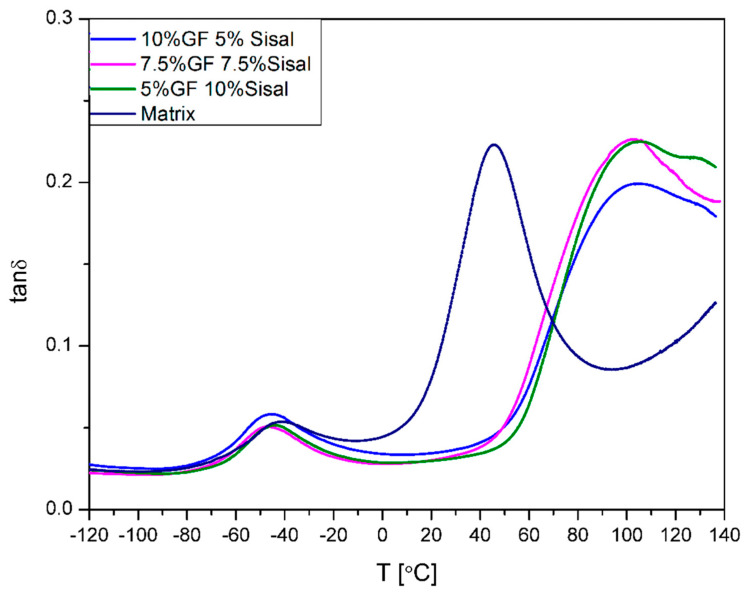
Tanδ values of obtained hybrids.

**Figure 14 materials-13-02028-f014:**
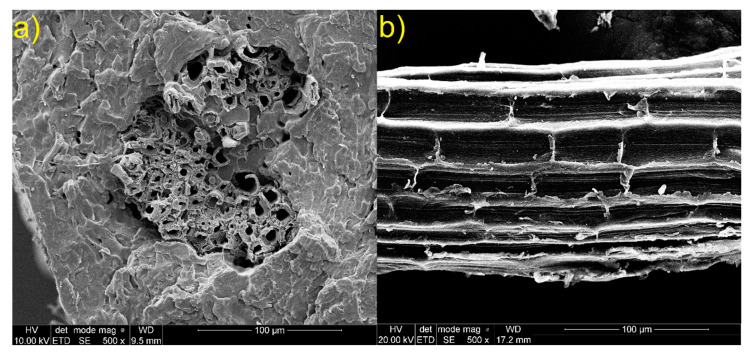
SEM images of sisal fiber (**a**) transversal cross-section of sisal in material; (**b**) longitudinal section of sisal.

**Figure 15 materials-13-02028-f015:**
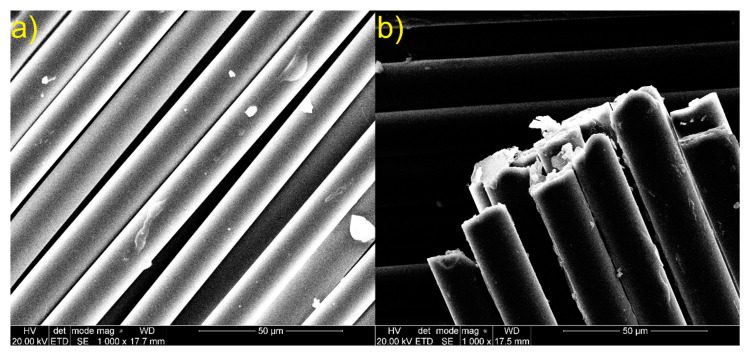
SEM images of glass fiber (**a**) lateral surface of glass fiber; (**b**) glass fiber ends.

**Figure 16 materials-13-02028-f016:**
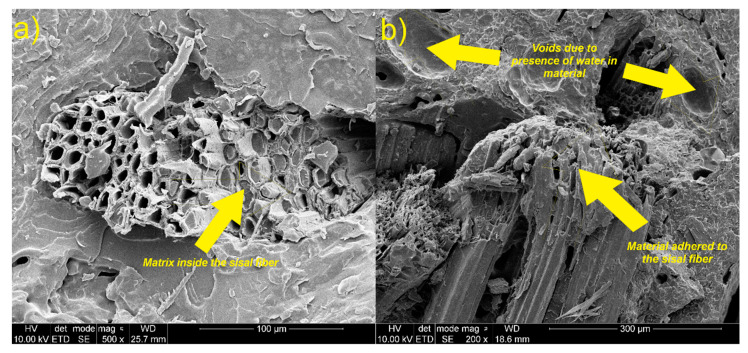
SEM images of sisal fiber in composite, (**a**) transversal crack of sisal fiber in the material; (**b**) the longitudinal crack of sisal fiber in the material.

**Figure 17 materials-13-02028-f017:**
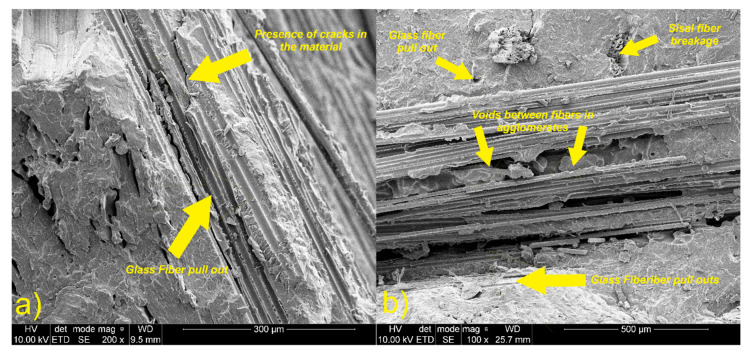
SEM images of glass fiber in composite, (**a**) group of defects in material; (**b**) agglomerate of glass fibers.

**Figure 18 materials-13-02028-f018:**
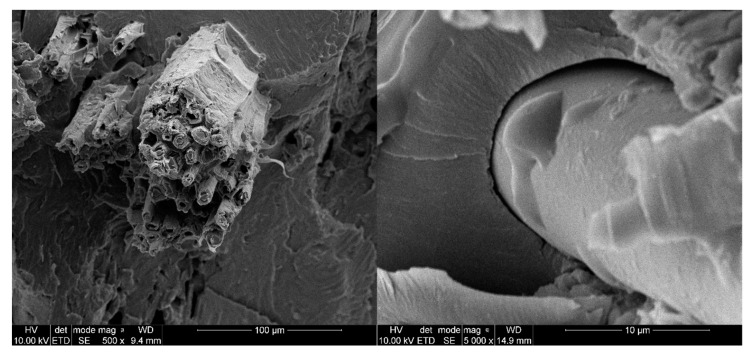
SEM images of glass and sisal fibers adhesion to matrix.

**Table 1 materials-13-02028-t001:** Nominal composition of obtained composites.

Sample	Content of Fiber	
	Glass Fiber	Sisal Fiber
**Matrix**	×	×
**5%GF**	5%	×
**15%GF**	15%	×
**5%SISAL**	×	5%
**15%SISAL**	×	15%
**5%GF10%SISAL**	5%	10%
**7.5%GF7.5%SISAL**	7.5%	7.5%
**10%GF5%SISAL**	10%	5%

**Table 2 materials-13-02028-t002:** Mechanical properties of obtained materials.

Sample	Flexural Strength (MPa)	Deformation (%)	Impact Strength (kJ/m^2^)	Hardness (Shore D)
**Matrix**	16.41 ± 0.51	12.62 ± 1.12	0.135 ± 0.004	56.6 ± 2.2
**5%GF**	21.93 ± 0.48	8.66 ± 0.78	0.169 ± 0.008	68.4 ± 2.1
**15%GF**	24.43 ± 0.71	8.9 ± 0.81	1.452 ± 0.051	70.8 ± 1.1
**5%SISAL**	23.80 ± 0.47	6.51 ± 0.63	0.240 ± 0.028	75.1 ± 1.8
**15%SISAL**	25.72 ± 0.53	5.27 ± 0.59	0.914 ± 0.128	73.9 ± 1.8
**5%GF10%SISAL**	34.97 ± 1.03	5.63 ± 0.43	1.074 ± 0.110	75.6 ± 1.5
**7.5%GF7.5%SISAL**	37.36 ± 1.13	5.05 ± 0.39	1.435 ± 0.073	75.7 ± 1.1
**10%GF5%SISAL**	43.04 ± 1.42	4.52 ± 0.47	1.699 ± 0.121	79.5 ± 1.6

**Table 3 materials-13-02028-t003:** Storage modulus at different temperatures and glass transition (T_g_) of hard and soft segments.

Sample	E’_20°C_ (MPa)	E’_60°C_ (MPa)	E’_100°C_ (MPa)	Soft Segments T_g_ (°C)	Hard Segments T_g_ (°C)
**Matrix**	584	141	76	−44.51	36.10
**5%GF**	1597	512	71	−45.21	46.27
**15%GF**	2369	809	138	−45.10	48.89
**5%SISAL**	1728	588	164	−44.82	45.46
**15%SISAL**	1583	731	329	−48.53	50.23
**5%GF10%SISAL**	1767	1358	362	−48.42	76.00
**7.5%GF7.5%SISAL**	1967	1290	307	−48.92	71.34
**10%GF5%SISAL**	2222	1700	569	−46.54	76.32

E’_xx°C_—storage modulus at different temperatures.

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
