# Peer review of "Bio-Based Polyurethane Composites and Hybrid Composites Containing a New Type of Bio-Polyol and Addition of Natural and Synthetic Fibers"

_materials, 2020, doi:10.3390/ma13092028_

Round 1
Reviewer 1 Report
Dear Authors,
in attachment You can find the file with my suggestions.

Author Response
Dear Reviewer
Thank you very much for the comments and tips contained in the review. We are very grateful because they helped us in improvement of this article. The changes indicated in the review are marked in blue in the manuscript. Below we are referring to Your comments:
- Line 10) Obtain is used to often. Rearrange the sentence
Thank you for pointing this out. Whole sentence was rearranged.
- Line 17) You have an increase of Tg for hard segments of PU for 40°C. This is not slight
Thank You very much for this comment. We agree with Your opinion that increase is not slight. We corrected this mistake.
- Line 30) The grammar of the sentence has been improved.
- Line 91) The grammar of the sentence has been improved.
- Line 96-98) technology or a process?
Thank you very much for pointing out this inaccuracy. Biomass liquefaction is a process.
- Line 123) The grammar of the sentence has been improved.
- Line 127,231) Biomass or Cellulose – use one term!
Thank You very much for this comment. You are right that we should use the same terminology in whole paper. Terminology was checked and corrected.
- Line 127) First you are talking about lignocellulose, then about the liquefaction process and then again about lignocellulose
Thank you very much for the comment. The text layout has been changed to be clearer.
- Line 129) Table 1. You mixed the sample name or samples?
Many thanks for indication of this oversight. We have made the necessary changes
- Line 141) Characterization of polyols, materials, composites and hybrids or maybe better only Characterization!
We changed title of this section.
- Line 147) The grammar of the sentence has been improved.
- Line 150) The grammar of the sentence has been improved.
- Line 151,170) This should be after rheology. It is good to follow the order as it is in results...
Thank you very much for pointing this out. The researchers are now correctly ordered.
- Line 183) Try to put this somewhere else, here you are talking about the hydroxyl number, and then in the middle of discussion - neutralization process!
Mentioned fragment was moved to the suitable place.
- Line 188,226) Image quality is bed! Check all image qualities in the manuscript!
Thank you for this comment. Figures were pasted again to the article to improve quality.
- Line 191) The grammar of the sentence has been improved.
- Line 194) there is no need for new paragraph!
The text was fitted to the previous paragraph.
- Line 196) After two hours at 150 °C the total amount of hydroxyl …..
Suggested comment was added to the sentence.
- Line 218-223,227) Please first comment one and then another, for instance: comment FTIR spectrum of glass fiber and then cellulose. you have here first GF, sisal fiber, then cellulose, then again sisal fiber, cellulose, then again lass fiber….
Many thanks for indication of this problem. The text layout has been changed to be clearer. The grammar of this description has been improved.
- Line 235) There is now FTIR analysis in this work! check FTIR analysis again! And Line) 273-286
Thank You for pointing out this issue. The description of FTIR analysis was checked. The text layout has been changed to be clearer. We checked FTIR bands and assigned them correctly. Moreover, we added few explanations.
- Line 247) The flow is linear!!! Polyol have characteristics of a Newtonian fluid. At small shear rates often you have some deviations! At the figure 6 is visible that at shear rates higher than 20 s-1 you have a flat line, there is no change in viscosity. According to this, the rheological calculation according to the model you should remove.
Thank You very much for this comment. We agree with the reviewer's opinion the bio-polyol exhibits Newtonian fluid characteristic. This has been changed in the manuscript. Our error resulted from the interpretation of all results, including those in the range of low shear rates. The reviewer is right that at low shear rates there are often deviations, which resulted in a poor interpretation of the results.
- Line 252-254) the loop is negligible!
We agree with the opinion of the bio-poly reviewer, the deviation of the flow curve hysteresis is small. This has been corrected in the manuscript.
- Line 288) only this? Tg, impact…???
We added necessary information.
- Line 293) in relation to the GF
With regard to the comment we added necessary relation.
- Line 296) The grammar of the sentence has been improved. We added few explanations.
- Line 303,313,321) The sentence/mistake was deleted.
- Line 336) hybrides. Delete samples 15% sisal and 15% GF, you have it on the figure 7!
The captions below the figures have been corrected. We have decided to delete sisal 15% and GF15% curves from the figures Fig. 8. Fig.10. and Fig.12.
- Line 349) Thank You very much for this comment. Suggested comment was added to the sentence.
- Line 359) The grammar of the sentence has been improved.
- Line 366) We exchanged places of 11. And Fig. 12.
- Line 381,382) The sentence/mistake was deleted.
- Line 391) We added few explanations to clarify the sentence. The grammar of the sentence has been improved.
- Line 394) mechanism of faliure: formation and spreding of cracks!
Thank you for the suggestion. An explanation has been added to the manuscript:
“Poor contact and bonding between this fiber and matrix resulted in poor stress transfer between phases. For this reason, fiber pull-off occurred and holes can be observed. Moreover, weak interactions between the polymer matrix and the fiber causes discontinuity in the composite structure, as a result of which microcracks are formed, which are responsible for further propagation of the fiber's cracks.”
- Line 395) Thank You very much for this suggestion. Suitable comment was added to the sentence.
- Line 405) This figure is not mentioned in the text. Delete the figure or comment the figure!
Thank you for pointing out this editorial error. Comment to this figure was accidentally deleted during text formatting. Paragraph was added to the text.
- Line 407) Suggested comment was added to the sentence. The grammar of the sentence has been improved.
- Line 417) Delete this sentence and write what is proved with FTIR?
Thank you for the suggestion. We changed conclusions in the new version of the manuscript.
- Line 423) Change according to my previous notes.
Thank you for the suggestion. It is changed in the new version of the manuscript.
- Line 426) You have increase in Tg for 40 °C of hard segment in hybrids. This is not a slight change.
We agree with Your opinion that increase is not slight. We corrected this mistake.
- Line 431) The grammar of the sentence has been improved.
- Line 432) has better adhesion to polymer matrix
Suggested comment was added to the sentence.

Reviewer 2 Report
The article is interesting, but requires a number of corrections before considering it for publication. I present my comments below:
- The authors did not provide affiliation
- Line 53 …psychical.. What are the psychical properties of sisal fiber?
- The description of the raw material suppliers should include the company name, city, state (if it is from the USA) and country.
- A scheme of biomass liquefaction reactions would be useful.
- How was the polyol with the best properties chosen? What does it mean that the polyol had the best properties? Please explain this.
- FTIR analysis does not allow to determine the chemical structure of new compounds, only the presence of characteristic functional groups e.g. OH. Authors should test the new polyol in 1H and 13C NMR spectroscopy.
- Authors should not mix description of polyol tests with composite tests. This should be separated.
- I understand that the authors did not consider sisal fibers as a reactive filler even though they had OH groups. I did not find information about it. The isocyanate index was changed in such systems. This affected the properties of the obtained composites.
- Fig 1 and Fig 2. Is 150C, 120C, 170C should be 120oC, 150oC, 170o
- 3 and Fig. 4 Indication appropriate functional groups in the FTIR spectrum would be desirable.
- Where is subsection 3.1? I did not find it.
- The conclusion section is not in the form required by Materials journal. This must be changed.
- Authors must re-edit the manuscript in accordance with the requirements of the Materials journal.
Author Response
Dear Reviewer
Thank you very much for the comments and tips contained in the review. We are very grateful because they helped us in improvement of this article. The changes indicated in the review are marked in red in the manuscript. Below we are referring to Your comments:
- The authors did not provide affiliation
Thank you for pointing out this oversight. Affiliations have been added to the article.
- Line 53 …psychical... What are the psychical properties of sisal fiber?
Thank you for pointing out this editorial error. It was caused by an inaccurate translation from Polish. In the following part of text, it should be mentioned only about mechanical properties.
- The description of the raw material suppliers should include the company name, city, state (if it is from the USA) and country.
We are really grateful for your valuable advice. All necessary information was added to the text.
- A scheme of biomass liquefaction reactions would be useful.
Thank you very much for pointing this out. The biomass liquefaction process is complicated. At the same time, biomass liquefaction and solvent condensation (which is used in excess) can occur. The product of the main reaction of cellulose liquefaction is levulinate, as shown in Fig. 1.
- How was the polyol with the best properties chosen? What does it mean that the polyol had the best properties? Please explain this.
Many thanks for indication of this inaccuracy. The best polyol was selected based on the following factors: hydroxyl number, biomass conversion, viscosity and water content
- FTIR analysis does not allow to determine the chemical structure of new compounds, only the presence of characteristic functional groups e.g. OH. Authors should test the new polyol in 1H and 13C NMR spectroscopy.
Again, Thank You for pointing out this issue. We are aware that the 1H and 13C NMR spectroscopy study would enrich our publication, but we have not decided to carry out this research for this article. Due to the complicated structure of the compounds obtained in the process of biomass liquefaction, testing of products will be the object of the next study, which we will want to publish in the next publication.
- Authors should not mix description of polyol tests with composite tests. This should be separated.
Thank You very much for this comment. You are right that these descriptions should be separated. These descriptions are currently in two different subsections – 3.1. and 3.2.
- I understand that the authors did not consider sisal fibers as a reactive filler even though they had OH groups. I did not find information about it. The isocyanate index was changed in such systems. This affected the properties of the obtained composites.
Thank you very much for pointing this out. During this work it was not assumed that sisal fiber can act as a reactive filler. We admit it was a mistake. Descriptions were added with information that due to its structure this type of fiber may be an active filler.
- Fig 1 and Fig 2. Is 150C, 120C, 170C should be 120oC, 150oC, 170oC
Thank you very much for the comment. The captions in the tables and figures have been corrected
- 3 and Fig. 4 Indication appropriate functional groups in the FTIR spectrum would be desirable.
Thank you very much for this advice. We marked most important signals on all figures.
- Where is subsection 3.1? I did not find it.
Thank you for pointing out this editorial error. Subsection 3.1. was accidentally deleted during text formatting. Subsection was added to the text.
- The conclusion section is not in the form required by Materials journal. This must be changed. Authors must re-edit the manuscript in accordance with the requirements of the Materials journal.
Thank you very much for pointing this out. The text has been reformatted to meet the journal's requirements. The conclusions were modified to acceptable form.

Round 2
Reviewer 2 Report
All my comments have been taken into account and the article has been significantly improved. I recommend publishing this article in the Materials journal.